# Effects of Sesame Oil Aroma on Mice after Exposure to Water Immersion Stress: Analysis of Behavior and Gene Expression in the Brain

**DOI:** 10.3390/molecules25245915

**Published:** 2020-12-14

**Authors:** Hiroaki Takemoto, Chiharu Take, Keito Kojima, Yamato Kuga, Tomoya Hamada, Tomoka Yasugi, Nanami Kato, Kazuo Koike, Yoshinori Masuo

**Affiliations:** 1Department of Pharmacognosy, Faculty of Pharmaceutical Sciences, Toho University, 2-2-1 Miyama, Funabashi, Chiba 274-8510, Japan; 1015223y@st.toho-u.jp (T.Y.); one.p-suikaaace77@docomo.ne.jp (N.K.); koike@phar.toho-u.ac.jp (K.K.); 2Laboratory of Neuroscience, Department of Biology, Faculty of Science, Toho University, 2-2-1 Miyama, Funabashi, Chiba 274-8510, Japan; daiouika0207@gmail.com (C.T.); keirobi0128@ezweb.ne.jp (K.K.); ticket_to_ride@outlook.jp (Y.K.); 5216071h@st.toho-u.jp (T.H.)

**Keywords:** sesame oil aroma, water immersion stress, antianxiety, elevated plus maze test, corticosterone, kruppel-like factor-4, dual-specificity phosphatase-1

## Abstract

(1) Background: Sesame has been popular as a healthy food since ancient times, and effects of the aroma component of roasted sesame are also expected. However, little research has been reported on its scent; (2) Methods: Jcl:ICR male mice were housed under water immersion stress for 24 h. Then, the scent of saline or sesame oil was inhaled to stress groups for 90 min. We investigated the effects of sesame oil aroma on the behavior and brains of mice; (3) Results: In an elevated plus maze test, the rate of entering to open arm and the staying time were decreased by the stress. These decrements were significantly enhanced by sesame oil aroma. Stress had a tendency to increase the serum corticosterone concentration, which was slightly decreased by the aroma. Expression of Kruppel-like factor-4 (*Klf-4*) and Dual-specificity phosphatase-1 (*Dusp-1*) in the striatum were increased by water immersion stress, and the level of *Klf-4* and *Dusp-1* in the striatum and hippocampus were significantly attenuated by sesame oil aroma (4) Conclusions: The present results strongly suggest that the odor component of sesame oil may have stress suppressing effects. Moreover, *Klf-4* and *Dusp-1* may be sensitive stress-responsive biomarkers.

## 1. Introduction

The effects of stress are inevitable while we live. All physical, chemical, biological and mental stimuli can be classed as stress, whether these involve eustress or distress. Moderate stress stimulates the sympathetic nervous system and promotes an improvement in judgment and behavioral abilities. However, excessive and chronic stress adversely burden the mind and body, and are concerned as risk factors causing disorders such as depression. Depression has a duration of more than a week with symptoms of the mind and body such as anxiety, hopeless feelings, reduced thinking ability, decreased interest and motivation, anorexia, and insomnia. Recently, the number of patients has increased in the world. Depression has a high risk of causing suicide impulses, and prevention, early diagnosis and treatment are important.

Suppression/elimination of stress is thought to be important for reducing the risk of developing depression. In recent years, various kinds of foods have drawn attention due to their stress relaxation effects. Sesame seeds have been popular as healthy foods since ancient times in Japan, and their imports are the second largest following the US and Europe [1]. Sesame contains antioxidants called sesame lignans [2]. It has been reported that active oxygen is eliminated by ingestion of vitamin E (α-tocopherol) and sesame lignan (sesamin, episesamin) which are abundantly present in sesame. These compounds are, therefore, thought to relieve fatigue and oxidative stress [3]. Sesame has also been suggested to have various effects such as antithrombotic effects [4], amelioration of lipid metabolism disorders [5], and protection of dopaminergic neurons [6].

As mentioned above, many studies have been reported on the effect of consuming sesame. However, there are few studies on the effect of the scent of sesame on the living body. The unique fragrance ingredients of roasted sesame seeds are composed of several molecules, such as pyrazine compounds with a fragrant aroma, furan compounds which possess a burning scent, etc. [7]. Pyrazines are also commonly found in the roasted aroma of coffee and tea [8]. It has been reported that coffee bean volatiles have anxiolytic and hypnotic effects in mice [9]. We previously demonstrated alterations in the expression of genes and proteins in the brain after stress, and suggested that the aroma of coffee beans has anti-stress effects on the brain [10]. Briefly, among the differentially expressed genes and proteins between the stress and stress with coffee group, NGFR (nerve growth factor receptor), trkC (tyrosine kinase receptor type 3), GIR (glucocorticoid-induced receptor), thiol-specific antioxidant protein, and heat shock 70 kDa protein 5 were identified as anti-oxidant or antistress factors.

The aim of the present study is to clarify the effect of sesame oil aroma on behavior and stress-responsive biomarkers (stress markers) in the blood and brain. Water immersion stress was applied to mice, which is thought to be suitable for preparing model animals before development of depression (undiagnosed condition). For behavioral analysis, we conducted an elevated plus maze test that is suitable for evaluating anxiety-related behaviors. We measured corticosterone—named as the stress hormone—levels in the serum and searched for stress marker candidates in the brain. Kruppel-like factor-4 (*Klf-4*) and dual-specificity phosphatase-1 (*Dusp-1*) are known to be negative regulators of the mitogen-activated protein kinase (MAPK) cascade, and the function of this cascade decreases in depression [11]. In this study, we focused on the expression *Klf-4* and *Dusp-1* genes in the hippocampus and striatum.

## 2. Results

### 2.1. Solid-Phase Micro Extraction (SPME)-GC-MS Analysis

Analyses of the volatile components of the sesame oil used in this study were performed on SPME-GC-MS. SPME is suitable for analyzing volatile compounds, adsorbing compounds in the headspace of the sample vial, and desorbing them directly into a GC injection port. A number of heterocyclic amines were detected from sesame oil aroma (Figure 1).

### 2.2. Body Weight

We first measured the body weight of mice for behavioral analysis. The rate of change in body weight before and after water immersion stress was −0.4% in the normal breeding groups (control/saline and control/aroma groups), and −8.2% in the stress groups (stress/saline and stress/aroma group). The decrease in body weight in the stress groups was significantly larger than that in the normal breeding groups (Figure 2). There were individuals with weight gain of 0.3 to 1.0% in the normal rearing groups, and weight loss was observed in all individuals in the stress-loaded groups.

### 2.3. Behavior

In an elevated plus maze test, the number of entries into the open arm and the closed arm was measured, and the ratio of the open arm entry number per entry number to the open and closed arms was calculated as the open arm entry rate (Figure 3). The average value of the open arm entry rate of each group was 48.7% in the control group, 29.5% in the stress group, 48.6% in the aroma group and 41.4% in the stress/aroma group (Figure 3A). The value of the stress group was significantly lower than the control group. In the stress/aroma group, a serious decrease was not observed, and a significant recovery was seen in comparison with the stress group. The mean value of the staying time of open arms in each group was 174.7 s in the control/saline group, 60.2 s in stress/saline group, 127.8 s in the control/aroma group and 133.3 s in the stress/aroma group (Figure 3B). Similar to the open arm entry rate, there was a significant reduction in the stress/saline group and a significant recovery in the stress/aroma group. This result suggests that the odor of sesame oil may have an anxiety depressing action. Meanwhile, the average value of the staying time of the closed arms of each group was 183.6 s in the control/saline group, 255.8 s in the stress/saline group, 246.6 s in the control/aroma group, and 252.6 s in the stress/aroma group (Figure 3C). In the three groups other than the control group, a slightly increasing tendency was observed, as compared with the control group, but no significant difference was observed between the groups. The average value of the total movement distance of each group was 25.5 m in the control/saline group, 16.1 m in the stress/saline group, 18.2 m in control/aroma group and 20.2 m in stress/aroma group (Figure 3D). A significant decrease was observed in the stress/saline group in the total movement distance, as compared with the control/saline group. These results indicate a decrease in locomotor activity due to the stress. In addition, although there was no significant alteration in the control/aroma and stress/aroma groups, a slight decreasing tendency was found. These results suggest that the aroma of sesame oil may also be involved in the reduction in locomotor activity.

### 2.4. Corticosterone Concentration

Significant differences were not observed among the four groups in regard to corticosterone concentration (*p* > 0.05, two-way ANOVA). However, the control/sesame group showed an increasing tendency compared to the control/saline group, and the stress/sesame group showed a tendency to decrease in comparison with stress/saline group (Figure 4).

### 2.5. Gene Expression

Absorbance of the obtained total RNA was measured using NanoDrop. The average A260/280 values in the four groups of striatum were 1.94 to 2.08, and the average A260/230 values were 2.03 to 2.19, respectively. The average A260/280 values in the hippocampus were 2.01 to 2.12, and the average A260/230 values were 2.08 to 2.19. Therefore, we considered that the purity of RNA extracted in the present study was sufficient. The expression level of *Klf-4* and *Dusp-1* was corrected by dividing by the value of GAPDH, one of housekeeping genes. The primer sequences of each gene are shown in Figure 5.

The expression level of *Klf-4* is shown in Figure 6. The expression level of *Klf-4* was significantly different among the four groups in the hippocampus and striatum (*p* < 0.05, two-way ANOVA). In the striatum, stress significantly increased the level of *Klf-4*. However, the stress/sesame group showed a significant decrease compared to stress/saline group (*p* < 0.05, Tukey’s test). The expression level of Dusp-1 is shown in Figure 7. The expression level of Dusp-1 was significantly different among the four groups in the hippocampus and striatum (*p* < 0.05, two-way ANOVA). In the hippocampus, the control/sesame and stress/sesame groups were significantly lower than the control/saline and stress/saline groups (*p* < 0.05, Tukey’s test). The similar differences were also found in the striatum with no significant differences between the stress/aroma and control/saline groups in this region. The stress/saline group was slightly augmented as compared to the control/saline in the striatum. However, there was not a significant difference between the two groups in either of the two regions.

## 3. Discussion

In the present study, we measured the body weight of mice before and after the 24 h loading of water immersion stress, and found that the rate of change in body weight after stress application was significantly decreased (Figure 2), indicating that stress sufficiently affected the living body. Glucocorticoid secretion is a key endocrine response to stress. Previous reports have suggested that glucocorticoid secreted due to acute stress has a lipolytic effect (lipolysis) and enhances the mitochondrial utilization of sugars, lipids and amino acids. Weight loss due to stress application is a supporting result of previous reports [12].

To evaluate the effect of stress on behavior, we conducted an elevated plus maze test (Figure 3). The stress significantly lowered the penetration rate to the open arm, the staying time at the open arms, and the total moving distance. The declining rate of entry into the open arm and staying time indicates that anxiety-like behavior was enhanced. In addition, the total migration distance is an indicator of spontaneous motor activity. Thus, these results suggest that the water immersion stress load in this study had an influence in suppressing the subsequent spontaneous motor activities. Inhalation of sesame oil scent significantly restored the penetration rate to the open arm and staying time at the open arm which were decreased by stress. These behavioral changes strongly suggest the possibility that the aroma of sesame oil suppresses anxiety induced by stress. On the other hand, the effect of stress on the total migration distance was not significantly suppressed by the scent of the sesame oil. However, the total migration distance in stress/sesame group was higher than the stress/saline group. Therefore, it is conceivable that the aroma component of sesame oil acts in the direction of increasing locomotor activity. Additionally, although it was not statistically significant, it is interesting that the behavior of control mice was also reduced by the scent of sesame oil. It is possible that the aroma components of sesame oil may contain some factor that suppresses spontaneous motor activity. As stated in the introduction, an anti-anxiety-like stress-reducing effect is recognized in coffee which has a fragrant component which is partially common to sesame seeds [9]. It is necessary to clarify in the future whether or not the fragrant component of sesame oil that suppressed spontaneous movement in this study is the same as the volatile component of coffee which produces a relaxing effect. 

From the GC-MS analysis of sesame oil aroma (Figure 1), it was found that heterocyclic amines and 2-methoxyphenol were common ingredients present in coffee [13]. The compound 2-methoxyphenol is used in traditional dental pulp sedation and it has been reported that 2-methoxyphenol is a potent scavenger of reactive oxygen radicals [14]. Furthermore, it has been reported that alkylpyrazine derivatives demonstrated a sleep prolongation effect. In particular, the duration of pentobarbital-induced sleep in mice was dose-dependently increased by 2,5-dimethylpyrazine, and the gamma-aminobutyric acid level in brains of mice was increased by the administration of 2,5-dimethylpyrazine [15]. The sedative and central inhibitory effect of these compounds may be involved in the stress relieving effect of the sesame oil aroma.

Sesame is rarely sniffed intentionally as is the case with an aroma oil. Generally we are most likely to be exposed to the scent of sesame in our daily life in the form of the fragrance of sesame itself, or when we eat dishes using sesame or sesame oil. Sesame seeds contain abundant antioxidants, and it has been clarified that removal of active oxygen alleviates oxidative stress [3]. The elevated plus maze test in this study suggested that the scent of sesame oil may have anxiolytic effects. Therefore, sesame seeds may be excellent anti-stress foods, having stress-suppressing effects not only after digestion but also in the process of sensing the aroma during eating.

Stress is transmitted to the hypothalamus via the cerebral cortex, and activates the “hypothalamus-sympathetic-adrenal medullary system (SAM)” axis and “hypothalamus-pituitary anterior lobe-adrenal cortex (HPA)” axis. When the HPA system is activated, glucocorticoids, such as cortisol and corticosterone, are released into the blood, which causes various physiological changes, such as increments in blood pressure, gluconeogenesis, cardiac contractility, cardiac output, and suppression of inflammation. However, hypersecretion of glucocorticoid is controlled via glucocorticoid receptors localized in the hippocampus, hypothalamus and pituitary gland (negative feedback). If glucocorticoid secretion persists due to excessive stress exposure, the hippocampal neurons are impaired and the glucocorticoid receptor decreases [16], which impairs hippocampal atrophy and the negative feedback mechanism, consequently causing depression-like behaviors [17]. In this study, the serum corticosterone concentration was highly variable among individuals, and barely any significant difference was observed among the four groups (Figure 4). However, the stress/saline group increased to about twice as much as the control/saline, which supports the idea that corticosterone is one of the stress response biomarkers. Since the corticosterone level in the control/sesame group was higher than the control/saline group, the scent of sesame seems to be stressful to the mouse itself. It is necessary to clarify whether stress due to scent is eustress or distress. However, the fact that the scent of sesame oil lowered the concentration of corticosterone under water immersion stress suggests that sesame may suppress the effect of distress. Moreover, although statistical significance was not observed, the scent of sesame oil showed a tendency to suppress the effect of water immersion stress. Regarding these effects of sesame oil, it is possible that significant differences may be observed by investigating the change after a stronger stress stimulus than water immersion stress.

We searched for stress marker candidates showing alterations in expression levels in both the brain and blood due to stress. In this study, we focused on *Klf-4* and *Dusp-1*. *Klf-4* is one of the four factors used for making induced pluripotent stem (iPS) cells, also called Yamanaka factor. It plays certain roles as a tumor suppressor for many kinds of cancers. However, the p53 tumor suppressor is inhibited by *Klf-4* in breast cancer, so *Klf-4* also functions as a tumor inducing factor [18]. It is also known that *Klf-4* affects various mechanisms of cells by inhibiting the MAPK pathway. In the present study, the stress caused a significant increase in *Klf-4* expression in the striatum and an increasing tendency in the hippocampus. These levels were significantly attenuated by the scent of sesame oil (Figure 6). Behavioral results suggest that the mouse used in this study was depressed by water immersion stress, and the MAPK pathway is considered to be showing reduced function. The odor component of sesame oil may inhibit the expression of *Klf-4* to repair the MAPK pathway.

*Dusp-1* is a member of a class of enzymes that remove phosphate groups from proteins. It is a major negative regulator of the MAPK cascade, an important signaling pathway involved in neuronal function [19]. It is known that the expression level of *Dusp-1* is elevated in the hippocampus of rats experiencing chronical stress [20] and decreased by treatment with antidepressants [21]. The induction of *Dusp-1* is not only a direct result of stress but also an important negative regulator of MAPK that contributes to the development of depressive symptoms. Previously, the significance of altered *Dusp-1* was tested in rodent models of depression and it was demonstrated that increased central *Dusp-1* expression, as a result of stress or viral-mediated gene transfer, causes depressive behaviors. Conversely, chronic antidepressant treatment normalizes the stress-induced *Dusp-1* expression and behavior, and mice lacking *Dusp-1* are resilient to stress [22]. Therefore, *Dusp-1* is representative of promising new drug targets for the treatment of depression and other mood disorders. We observed that the scent of sesame oil significantly inhibited the expression level of *Dusp-1* in both the hippocampus and striatum of the control and stress groups (Figure 7).

In the present study, *Dusp-1* expression in the hippocampus was not increased due to water immersion stress, but we found an increasing trend in the striatum. The intensity of water immersion stress in this study may be too weak to cause a significant increase in *Dusp-1* expression. However, suppression of *Dusp-1* expression under normal and stress conditions suggests that the scent of sesame may suppress stress, not only in water immersion stress, but also in normal environment.

In this study, a significant anxious-like behavior was counteracted by the scent of sesame oil. Changes in serum corticosterone levels due to stress were greatly varied by individuals and alterations were not statistically significant. However, its levels were increased by stress, and tended to be suppressed by the scent of sesame. The expression level of *Klf-4* in the striatum was significantly increased by stress, which was significantly inhibited by the sesame oil aroma. Similar changes were found in *Klf-4* expression in the hippocampus. *Dusp-1* expression in the striatum and hippocampus showed alterations similar to those of *Klf-4*. However, sesame oil significantly inhibited the expression levels of *Dusp-1* in both regions, regardless of the presence or absence of stress exposure. 

Our results suggest that the aroma component of sesame oil has a stress-suppressing effect in behavior via alterations in the expression of *Klf-4* and *Dusp-1* in the brain. Further studies on the interaction of different kinds of stress could be necessary. The present study is the first step to link the changes in the level of expression of intracerebral factors and in behavior due to the scent of sesame.

## 4. Materials and Methods 

### 4.1. Materials

Aromatic sesame oil used in this study was purchased from Kuki Sangyo Co., Ltd. (Mie, Japan). This oil was produced by pressing and extracting from white sesame from Guatemala, followed by a cooling process.

### 4.2. Volatile Compound Analysis

Samples (1 g) were placed in a glass vial (10 mL) and heated at 40 °C for 5 min. Released volatiles were adsorbed on SPME fiber polydimethylsiloxane/carboxen/divinylbenzene (PDMS/Carboxen/DVB, Sigma-Aldrich Corp., St. Louis, MO, USA) at 40 °C for 20 min. The volatiles adsorbed on SPME were separated using an Agilent 7890B gas chromatograph (Agilent Technologies Inc., Santa Clara, CA, USA) with a fused silica capillary column (DB-WAX, 30 m × 0.25 mm × 0.25 μm; Agilent Technologies Inc.). Volatiles isolated were analyzed using an Agilent 5977A mass spectrometer (Agilent Technologies Inc.). Operating conditions were as follows: injector temperature, 250 °C; helium flow rate, 1 mL/min; oven temperature, 35 °C for 5 min and then programmed to increase at 5 °C/min to 120 °C, and increased at 15 °C/min to 220 °C and held for 6 min. Mass spectra were obtained by EI ionization at 70 eV over 29 to 290 mass units, with an ion source temperature of 230 °C. Volatile compounds were identified by comparison of their mass spectra similarities with standard compounds and the NIST 02 spectral library of the gas chromatography-mass spectrometry (GC-MS).

### 4.3. Animal Experiment

This study was carried out according to the animal experiment handling provision by Toho University Animal Experiment Committee. Animal experiment design plan has been approved by the committee. Jcl:ICR male mice at 5 weeks of age purchased from Clea Japan (Tokyo, Japan) were housed in an acrylic cage, three animals per cage, and kept under a 12 h light/dark cycle (light period 08:00–20:00) at 24 ± 2 °C. Food and water were allowed ad libitum. After adapting to the rearing environment over a week, mice were randomly divided into 4 groups of 6 mice: control/saline (normal rearing and saline inhalation) group, stress/saline (stress load and saline inhalation) group, control/sesame (normal breeding and sesame oil aroma inhalation) group, and stress/sesame (stress load and sesame oil aroma inhalation) group. The Stress/saline and stress/sesame groups received water immersion stress. These animals were bred for 24 h in a cage with water in the limbs of mice were immersed to varying degrees (water depth of about 1 cm). This stresses the mice due to discomfort and insomnia induction. Meanwhile, control/saline and control/sesame groups were bred under normal conditions with wood chips in a cage. Since the manure odor of the mouse during the water immersion stress is strong, it may influence the aroma inhalation later. Therefore, cages were put inside the draft (fume hood) to minimize manure odor during the stress loading. For control groups, breeding was carried out in the same draft for 24 h to unify the breeding environment. Inhalation was started from 10:30 just after 24 h of stress loading or normal rearing. The control/saline and stress/saline groups were exposed to saline resulting in them inhaling it, and the control/sesame and stress/sesame groups instead inhaled the sesame oil aroma for 90 min. The odor was placed in a glass box in which a piece of filter paper impregnated with 50 μL of physiological saline or sesame oil was adhered to the inside. This study was carried out according to the animal experiment handling provision by Toho University Animal Experiment Committee. Ethic Committee approval numbers were Dosho 16-52-292, 17-53-292, 18-54-292, 19-55-292, and 20-21-450.

### 4.4. Weight Measurement

In order to check whether water immersion for 24 h was stressful to the mouse, body weight was measured before and after stress application, and the rate of change was calculated. Measurements before stress were carried out when the mice were randomly divided into 4 groups. Measurements after stress were performed after each aroma inhalation, since moisture was contained in the body hair of mice due to water immersion.

### 4.5. Behavioral Analysis

Evaluation of anxiety-like behavior was performed by the elevated plus maze test for 10 min just after completion of inhalation of the aroma. The maze consisted of an arm length of 30 cm, arm width of 5 cm, height from the floor of 60 cm, and closed arm wall height of 20 cm. The test was carried out in a dim room with an illuminance of 140 ± 10 lux. Immediately after putting the mouse in the center of the maze, the test room was unattended. The behavior of animals was captured with a video camera on the ceiling. The total moving distance and average moving speed were automatically recorded using behavioral tracking software ANY-maze (Muromachi Kikai, Tokyo, Japan). Regarding the number entering into the open arm and the staying time, the number entering into the closed arm and the staying time, the video image was observed and the average value was calculated by measuring twice for each individual.

### 4.6. Dissection

After 24 h under the conditions described above, the mice were decapitated from 12:00, and the blood and whole brain were collected. The blood was centrifuged (2000× *g*/4 °C, 10 min), and the serum was dispensed into another 2 mL tube and stored at −80 °C. The whole brain was dissected on ice, and the striatum and hippocampus were sampled. The brain tissue was put in a 2 mL tube, immediately frozen in liquid nitrogen, and stored at −80 °C. Thereafter, the frozen tissue was pulverized in liquid nitrogen to make it into powder form. Powdery tissue was divided into 2 mL tubes and stored at −80 °C.

### 4.7. Corticosterone Concentration

Serum corticosterone concentration was measured using the Corticosterone Enzyme Immunoassay Kit (Arbor Assays, Ann Arbor, MI, USA). In each tube, 5 μL of dissociation reagent and 5 μL of serum were mixed and incubated at room temperature for 5 min or more. The serum was diluted 100 times by adding 490 μL of assay buffer. The procedure was completed according to the manufacturer’s protocol. Each reaction was carried out on a 96-well plate, and the absorbance at 450 nm was measured with a plate reader. Corticosterone concentration was calculated using My Assay (https://www.myassays.com/home.aspx).

### 4.8. Total RNA Extraction

One mL of Qiazol (Qiagen, Hilden, Germany) was added to each brain tissue sample and stirred until the sample was completely dissolved. Total RNA was extracted with RNeasy Mini Kit (Qiagen) and RNase-Free DNase set (Qiagen) with the manufacturer’s protocols. RNA was dissolved in RNase-free water, and the concentration and the purity of the obtained total RNA was measured using NanoDrop. Samples with low purity were ethanol precipitated with Ethachinmate (Nippon Jean, Tokyo, Japan). Total RNA was stored at −80 °C.

### 4.9. cDNA Synthesis

Reverse transcription was performed from the obtained RNA to synthesize cDNA using ReverTra Ace^®^ qPCR RT Master Mix (Toyobo, Osaka, Japan). First, the amount of sample was calculated so that the RNA concentration of each sample was 1 pg/μL to 1 μg/μL in 10 μL. To 2 μL of each sample, 6 μL of nuclease-free water was added, and 2 μL of 2 × RT Master Mix (ReverTra Ace^®^ qPCR RT Master Mix, Toyobo) was added. cDNA was synthesized by reverse transcription using GeneAmp PCR System 9700 (Applied Biosystems, Thermo Fisher Scientific). Conditions for cDNA synthesis were 15 min at 37 °C, 5 min at 50 °C and 5 min at 98 °C. The synthesized cDNA was stored at −20 °C. 

### 4.10. Real Time Reverse Transcription-Polymerase Chain Reaction (Real Time RT-PCR)

For real time RT-PCR, we used TB Green Premix Ex Taq II (Takara Bio, Shiga, Japan). To 2 μL of each cDNA, 1.6 μL of primer (Forward and Reverse, 0.8 each) (Figure 5) (Takara Bio) was added, and 8.5 μL of RNase free water, 12.5 μL of TB Green *Premix Ex Taq* II and 0.4 μL of Rox Reference Dye was added to make a total of 25 μL. The expression level of cDNA was measured using an Applied Biosystems^®^ 7500 real-time PCR system (Thermo Fisher Scientific). Holding Stage was 50 °C for 2 min and 95 °C for 10 min. Cycling Stage for 40 cycles was 95 °C for 15 s and 60 °C for 1 min. Melt Curve Stage was 95 °C for 15 s, 60 °C for 1 min, 95 °C for 30 s, and 60 °C for 15 s. Analysis of the gene expression level of each sample was carried out in comparison with the expression level of glyceraldehyde 3-phosphate dehydrogenase (*GAPDH*) which is a housekeeping gene. Expression levels of *Klf-4* and *Dusp-1* were calculated as correction values divided by the expression level of *GAPDH*. 

### 4.11. Statistical Analysis

Results were expressed as mean ± SEM. For the significance among 4 groups, ANOVA analysis was performed and further analysis was conducted using the Tukey’s test method. In some cases, the results were analyzed by Student’s *t*-test to determine significant difference between two groups. *p* < 0.05 was considered to be statistically significant.

## 5. Conclusions

The effect of sesame oil aroma on behavior and stress-related biomarkers was investigated. Water immersion stress for 24 h induced a state of anxiety and increased serum corticosterone levels and expression of kruppel-like factor-4 in the striatum and dual-specificity phosphatase-1 in the hippocampus. However, the sesame oil aroma suppressed the above-mentioned stress-induced changes. The results of the present study suggest that the sesame oil aroma could be useful as an anti-stress agent for the treatment of various mental disorders. 

## Figures and Tables

**Figure 1 molecules-25-05915-f001:**
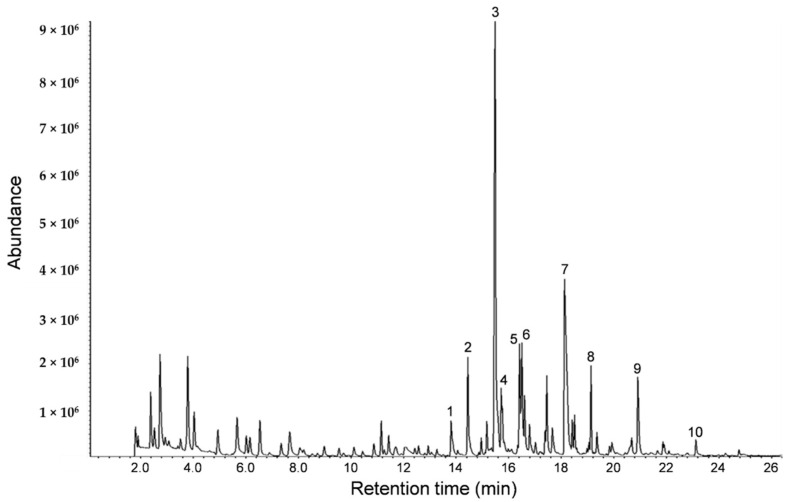
A GC chromatogram of the SPME-GC-MS analysis of the sesame oil. The retention time (RT) and percent of composition of the ten peaks identified in this study are indicated below. 1: pyridine, RT 13.8 min, 1.7%; 2: pyrazine, RT 14.5 min, 5.6%; 3: 2-methylpyrazine, RT 15.5 min, 26.5%; 4: 4-methylthiazole, RT 15.7 min, 2.0%; 5: 2,5-dimethylpyrazine, RT 16.4 min, 4.0%; 6: 2,6-dimethylpyrazine, RT 16.5 min, 4.4%; 7: acetic acid, RT 18.1 min, 10.4%; 8: pyrrole, RT 19.1 min, 2.7%; 9: 3-furanmethanol, RT 20.9 min, 1.9% and 10: 2-methoxyphenol, RT 23.1 min, 0.5%.

**Figure 2 molecules-25-05915-f002:**
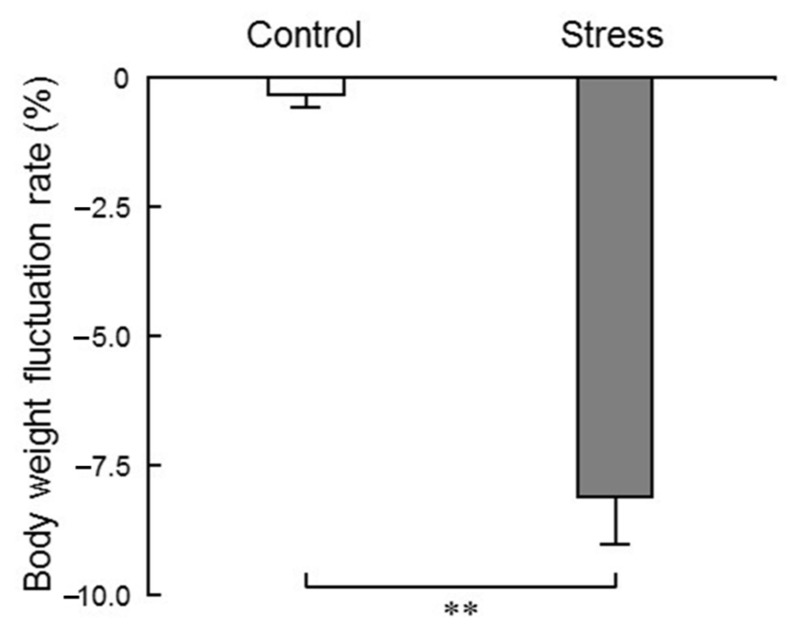
The fluctuation of body weight. Results represent mean ± SEM (n = 6). ** *p* < 0.01 (two-way ANOVA followed by Tukey’s test or the Student’s *t*-test).

**Figure 3 molecules-25-05915-f003:**
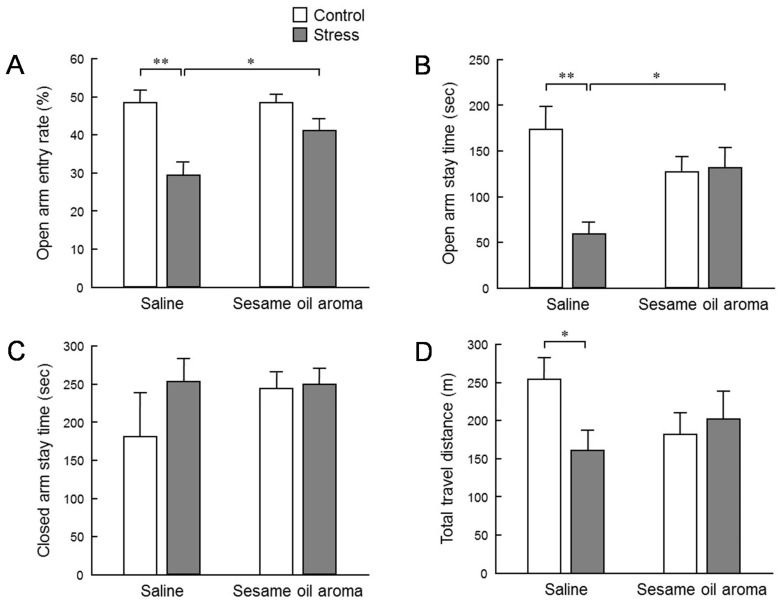
Elevated plus maze test. (**A**): open arm entry rate (%), (**B**): open arm stay time (sec), (**C**): closed arm stay time (sec), (**D**): total travel distance (m). Results represent mean ± SEM (n = 6). * *p* < 0.05, ** *p* < 0.01 (two-way ANOVA followed by Tukey’s test or Student’s *t*-test).

**Figure 4 molecules-25-05915-f004:**
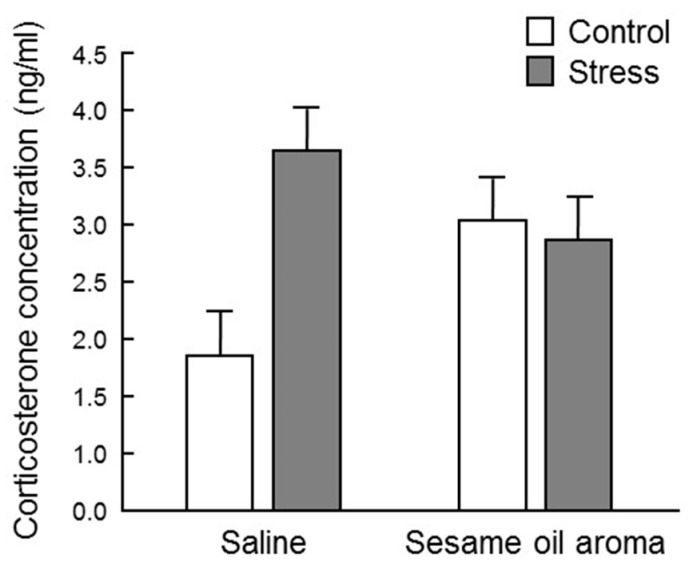
Serum corticosterone concentration. Results show mean ± SEM (control/saline, stress/saline, control/sesame, stress/sesame, n = 6).

**Figure 5 molecules-25-05915-f005:**
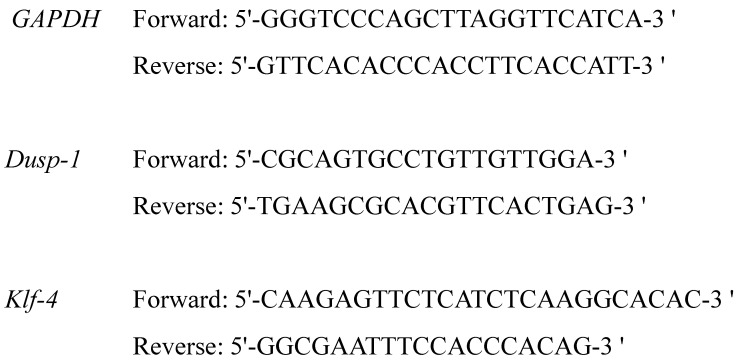
The primer sequence of each gene measured in the present study.

**Figure 6 molecules-25-05915-f006:**
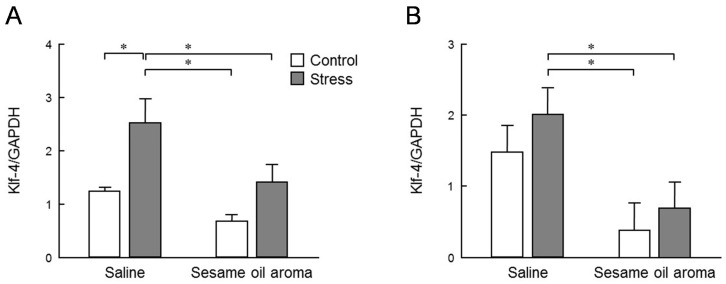
Expression level of *Klf-4* in the striatum (**A**) and hippocampus (**B**). Results show means ± SEM (control/saline, stress/saline, control/sesame, stress/sesame, n = 6). * *p* < 0.05 (two-way ANOVA followed by Tukey’s test or Student’s *t*-test).

**Figure 7 molecules-25-05915-f007:**
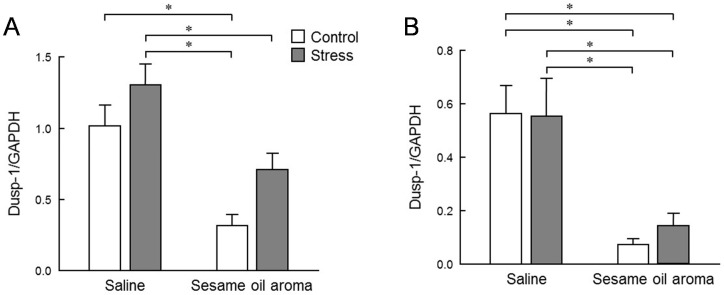
Expression level of *Dusp-1* in the striatum (**A**) and hippocampus (**B**). Results show mean ± SEM (control/saline, stress/saline, control/sesame, stress/sesame, n = 6). * *p* < 0.05 (two-way ANOVA followed by Tukey’s test or Student’s *t*-test).

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
