# Peer review of "Effects of Sesame Oil Aroma on Mice after Exposure to Water Immersion Stress: Analysis of Behavior and Gene Expression in the Brain"

_molecules, 2020, doi:10.3390/molecules25245915_

Round 1

Reviewer 1 Report

In the present manuscript, the authors study the effect of sesame oil aroma on behaviour and stress-responsive biomarkers in the blood and brain of ICR male mice.

The authors may address the following issues to strengthen the manuscript:

  1. In the introduction, the authors indicate that sesame oil is produced by pressing and extracting from white sesame from Guatemala, followed by a cooling process. However, in the Materials and Methods section, it is indicated that the Aromatic Sesame Oil was purchased from the company Kuki Sangyo (Yokkaichi, Japan). It is necessary to clarify this point.
  2. It is necessary to include at least the qualitative and quantitative chemical analysis of the sesame oil used.

Reviewer 2 Report

Interesting MS and topic. Some suggestions:

Introduction part:

"We previously demonstrated alterations in the expression of genes and proteins in the brain after stress" A review paragraph about expression of genes -> proteins -> stress is needed to focus the reader on the investigation.

Sesame oil extract preparation should be described as a paragraph in the methodology; the introduction part should avoid explanation.

"Therefore, it is important to analyze the fragrance component of sesame oil by GC/MS and to examine behavioral scientific changes and physiological effects when inhaling each aroma component " I think there are some interesting and recent works about it that should be found, see Phytotherapy journal and Journal of ethnoparmacology, and cited here. 

A conclusion paragraph is needed.

Try to associate increase levels of DUSP-1 and Klf-4 with concrete increased final products of the MAPK route, the real in-vivo effects and it relation with stress and disease. Some of them are cited but disperse over the manuscript and should be collected prior the conclusions. 

Round 2

Reviewer 1 Report

I appreciate your conscientious efforts to revise and improve your manuscript. I think it will make a very nice contribution to the field.